# The Internal Determinants of Gender Diversity and Its Non-Linear Impact on Firms' Performance: Evidence from the Listed Companies in Palestine Exchange

Abdelrahman J. K. Alfar [1,*] , Nariman Abuatwan [2] , Mohamed Elheddad [3] and Mohammad Qaki [4]

1 Global Business School for Health, University College London (UCL), London WC1E 6BT, UK
2 Administrative and Financial Business Department, Palestine Technical University Kadoorie, Yafa Street, Tulkarm P.O. Box 7, Palestine
3 Teesside University International Business School, Teesside University, Middlesbrough TS1 3BX, UK
4 Department of Finance and Banking, Birzeit University, Birzeit P.O. Box 14, Palestine
* Correspondence: a.alfar@ucl.ac.uk

**Abstract:** This study mainly aims to test the impact of gender diversity on a firm's performance. Namely, the non-linear and the quantile impact on the listed companies in Palestine Exchange during the period 2010 to 2020. The study also aims to determine the impact of a firm's internal characteristics on gender diversity. The study uses instrument analysis, traditional panel models, and quantile regression to fulfil the aims. The results demonstrate the existence of a critical mass for the impact of gender diversity on firms' performance and that mass is about 30% for the ROA and 41% for the EPS.

**Keywords:** gender diversity; firm performance; non-linear impact

## 1. Introduction

Following contemporary corporate financial crises and scandals in this era of inclusion and equality, a vital issue has been raised: would we gain different outcomes if more females led corporations in the United States and the rest of the world (Adams and Funk 2012)? There are several motives to deem that the answer is yes. Women executives are more thoughtful than males when making critical company decisions (Huang and Kisgen 2013; Levi et al. 2014). Because board composition is so important in corporate governance, the presence of female directors in the boardroom must have a substantial impact. Daily et al. (1999) argue that choosing females as directors is a hands-on need; having women on corporate boards increases the diversity of viewpoints. According to Adams and Ferreira (2009) and Gul et al. (2008), female board members are more vigilant monitors and require more audit efforts than male board members. Francoeur et al. (2008), who backed such initiatives, stressed the relevance of policies such as these in promoting women's success in business. These initiatives, according to Adams and Ferreira (2009), are founded on the belief that women on boards may have a substantial impact on corporate governance. Firms with females on their boards have long been concerned with greater corporate governance and ethical behaviour, according to a Catalyst (2013) analysis. As a result, various countries have enacted legislation to ensure that women are represented on boards of directors. Regulators and society are pushing women to join previously male-dominated professional areas. Even though the number of female employees at various levels of the organizational structure is increasing every day, women are seen to be underrepresented in top-level managerial roles and on corporate boards all over the world. Women made up 19.2 percent of corporate boards in the US, 22.8 percent in the UK, 18.2 percent in Spain, 9.5 percent in India, and 3.1 percent in Japan (Catalyst 2013). The majority of these initiatives take the form of quotas, time periods, and noncompliance fines. The government of Norway was the first to implement a 40 percent female quota in 2003,

and Spain followed suit in 2007, Belgium, France, Finland, Iceland, India, Italy, and Kenya are among the countries that have joined the bandwagon. With these arguments in mind, many European countries are promoting, if not demanding, the addition of more women to corporate boards of directors. Can a more diversified board of directors, however, lead to better company performance?

Many scholars have investigated the impact of female directors on corporate performance in recent years in an attempt to answer this question. However, the existing empirical evidence is equivocal, as the majority of studies focus on enterprises in the United States and a few other developed economies. Adams and Ferreira (2009), for example, show a negative relationship between board gender diversity and business performance in the United States, largely attributing the unfavourable relationship to women directors' over-monitoring. Furthermore, according to Bøhren and Staubo (2014) and Ahern and Dittmar (2012), The findings are attributed to the idea that the rule forces companies to choose younger, less experienced women as directors and ineffective boards of directors. Carter et al. (2003) and Campbell and Mínguez-Vera (2008), however, find a strong and positive relationship between the percentage of women on boards of directors and firm performance. According to Triana et al. (2014), board gender diversity can either drive or inhibit strategic improvements, depending on the firm's performance and the authority of women directors. Other research in the United States and Denmark has not shown a substantial link between board gender diversity and firm performance (Farrell and Hersch 2005; Carter et al. 2010; Rose 2007).

In the context of Palestine, the aforementioned conclusions do not provide a clear direction. Palestine's corporate governance is much worse than that of the United States and other wealthy countries. Is it true that having a woman as a director boosts a company's performance in Palestine?

The study aims to test the impact of gender diversity on firms' performance. Namely, the non-linear and the quantile impact at the listed companies in Palestine Exchange during the period from 2010 to 2020, besides testing the impact of firms' internal characteristics on gender diversity.

The main contributions of this study are three-fold. Firstly, to test the existence of the critical mass impact of female representation in PEX BoD. Secondly, to determine the impact of firms' internal characteristics on gender diversity in PEX. Thirdly, to test at which level of output the impact of gender diversity on firms' performance is more effective.

The remainder of the paper is laid out as follows: literature review; data collection; methodology; results and discussion; and finally, the conclusion.

## 2. Literature Review

This section reviews the relevant theories on board gender diversity and its impact on firms' performance. Gender diversity on corporate boards (board gender diversity) research is a decent topic that continues to receive attention. As a result of the low participation of women on corporate boards, numerous nations have adopted gender quota laws mandating the nomination of women to corporate boards. Substantial research on board gender diversity has been conducted in particular during the continuous increase in the number of female directors. In this section, the study investigates how the strands of literature on board gender diversity have evolved in areas such as corporate governance and finance. Therefore, the study focuses on the studies that shed the light on the representation of gender diversity on corporate boards and how this affected its performance and value.

The majority of the work on board gender diversity is empirical and draws on management literature for its theoretical foundations. Pioneer research on board gender diversity has emphasized theories in management at three levels: firm, board, and the individual level. Human capital theory, social network and social cohesion theory, social identity theory, agency theory, resource dependency theory, and token status theory are some of the most commonly referenced theories. Human capital theory (Becker 1985) looks at how an individual's cumulative store of skills, education, and experience affects the

development of productive and cognitive capabilities, which benefits both the individual and the organization. The diversified and unique human capital of a corporate board is considered a crucial resource for the firm in the context of corporate governance. Social Identity Theory (Tajfel and Turner 1979) is one such theory that recognizes how the presence of women alters group dynamics and leads to heterogeneity in decision-making. Turner and Tajfel's (1986) social identity theory investigates the impact of group membership on an individual's identity, including gender, race, class, and occupation. These identities establish group boundaries and may result in higher ratings of ingroup members, strengthening the outgroup's greater admission barriers. A male-dominated board, for example, may enhance group boundaries by excluding women from board directorship. Upper Echelons Theory is another supporting theory that recognizes the relevance of women in strategic decision-making in a company (Hambrick and Mason 1984). However, the resource dependency theory, token status theory, sex-role stereotypes and agency theory are the most common management theories discussed in the literature of board gender diversity since the organization structure is geared at collecting and exploiting resources via contracts (Berle and Means 1932).

### 2.1. Resource Dependence Theory and Board Gender Diversity

The principle of resource dependence asserts that enterprises rely on the resources in their surrounding environments to exist. Businesses are in danger as a result of these dependencies. Businesses can cultivate links with the external bodies that govern those resources to lessen reliance and their associated uncertainties. (Pfeffer and Salancik 1978)

Corporate board links provide three benefits, according to Pfeffer and Salancik (1978): advice and counsel, legitimacy, and communication routes. In terms of advice and counsel, Kravitz (2003) and Huse and Solberg (2006) suggest that gender-diverse boards are associated with higher-quality board debates of complicated topics, some of which may be unpalatable to all-male boards.

As for legitimacy, firms' practices are legitimized by respecting societal norms and values. The "value-in-diversity" hypothesis, proposed by Cox et al. (1991), states that as women's equal rights become more mainstream in society, corporations gain legitimacy by selecting female directors to their boards.

With regard to communication channels, women CEOs are more suited to link their companies to female customers, women in the workforce, and society at large due to their diverse life experiences and opinions. Hillman et al. (2007) use resource dependence theory to study board gender diversity and discover that enterprises with gender-diverse boards can reap these benefits in the United States. In conclusion, the resource dependence theory suggests that gender-diverse boards have positive impacts.

### 2.2. Agency Theory, Corporate Governance and Board Gender Diversity

When managers make company decisions, agency difficulties arise when they do not consider the best interests of shareholders. Enhancing corporate board oversight is one solution. According to Fama and Jensen (1983), effective board advice and oversight are critical in resolving these conflicts of interest. Women directors are more involved in monitoring activities, according to empirical research. Gul et al. (2008) and Adams and Ferreira (2009), for example, illustrate that more gender-diverse boards necessitate increased auditing and managerial accountability. As a result, the conflict of interest between management and stockholders decreased and business performance increased.

The impact of board gender diversity on corporate choices is also influenced by the quality of a company's governance. Adams and Ferreira (2009) argue that excessive over-monitoring that results from board gender diversity can be detrimental to firm value in well-governed corporations. Gul et al. (2011), however, argue that having a gender-diverse board can help companies improve their governance.

Allen et al. (2005) state that emerging markets' legal institutions in terms of investor protection, corporate governance, accounting standards, and government quality are far



less developed than those in the other industrialized nations such as the United States. As a result, we expect that over-monitoring is unlikely to be an issue in the case of Palestine. Due to the aforementioned partial substitution effect, gender-diverse boards may have a positive impact on business performance under Palestine's current condition of poor corporate governance.

*2.3. Token Status Theory (Critical Mass Theory), Sex-Role Stereotypes, and Board Gender Diversity*

Kanter (1977) refers to women or minorities in high management as "tokens" or "solos," referring to someone who is the lone representative of a demographic group (e.g., gender and race). Kanter (1977) goes on to say that onlookers tend to misrepresent female token managers' images in ways that are more directly tied to femininity than to leadership skills. This distortion of the image leads to sex-role stereotypes of female directors, which are at odds with public opinions of leaders. According to Kulich et al. (2007), the salary discrepancy between male and female executive directors is exacerbated by gender-role stereotypes of women executives. Similarly, the study on schemas implies that people form mental models of particular work characteristics (Lee and James 2007). Male job applicants are more likely than female job applicants to be associated with those traits because men hold the majority of senior management positions (Powell and Butterfield 2002). Women's historical symbolic status in top management further fosters preconceptions that women lack the required qualifications for such jobs (Lee and James 2007).

A lone female director may be considered as a mere "token" by both internal and external stakeholders due to the token status and sex-role stereotypes of female directors, and her impact on corporate decisions is likely to be limited. According to the critical mass theory developed by Kanter (1977), critical mass arrives at 30 percent female directors (Strydom et al. 2016). The critical mass theory on board gender diversity is an extension of the token status theory, positing that "one is a token, two is a presence, and three is a voice" (Kristie 2011). "The magic seems to happen when three or more women serve on a board together" concluded by Kramer et al. (2006) who expected that having three or more women on a board of directors can produce a critical mass where women are no longer perceived as outsiders and can have an impact. It acknowledges the unique viewpoint and skills that women can contribute to a company. The Catalyst Information Centre, which supports this endeavour, discovered that boards with three or more female directors performed better financially than boards with fewer women on the board of directors (Catalyst 2013).

Token status theory can be applied to the study of gender diversity in the board of directors of a company by examining the perceived value and utility of having women represented on the board. According to token status theory, the status of a token (in this case, the token being a woman on the board) is determined by its perceived utility, rarity, and liquidity (Kramer et al. 2006).

In terms of utility, the presence of women on the board may be seen as beneficial to the company because of the diverse perspectives and experiences they bring, which can improve decision-making and overall performance.

In terms of rarity, the proportion of women on the board may be seen as a measure of the company's commitment to gender diversity. A company with a high proportion of women on the board may be seen as more progressive and forward-thinking than a company with a low proportion of women on the board.

In terms of liquidity, the ease with which women can be recruited and retained for board positions may be a factor in their perceived value. A company with a track record of successfully recruiting and retaining women on the board may be seen as more attractive to potential investors and stakeholders than a company with difficulty in this area.

By applying token status theory to the study of gender diversity in the board of directors, researchers can examine the factors that influence the perceived value and utility

of having women represented on the board, and how these factors may impact the overall performance and success of the company.

Brahma et al. (2021) studied gender diversity, female traits, and the financial performance of UK FTSE 100 enterprises in a recent study. This study reveals a positive and significant association between gender diversity and business performance using critical mass theory to measure boardroom female representation. When three or more women are appointed to the board, the results are clear and important. Female age, education, and executive director status significantly affect post-appointment financial success. After controlling for endogeneity and using return on assets and Tobin's Q as performance indicators, the results remain the same. Garanina and Muravyev (2021) in their new study related to the Russian context, found that when at least three women are assigned to corporate boards, there is a favourable influence of gender diversity on performance; but, when just one woman is nominated to the board, these same findings do not hold true. The most recent research on the subject in the context of Portugal, and the findings indicate that female presence is positively correlated with ROA when there are at least two women on the Board or when the percentage of women is at least 20% (Carmo et al. 2022).

## 3. Gender Diversity on Corporate Boards Has an Impact on Company Performance

Researchers throughout the world are currently focusing on assessing the influence of women directors on firms and other associated features in light of these theories and regulatory actions in creating equality on corporate boards. The majority of previous research on gender diversity on corporate boards has focused on how board diversity affects business performance. Carter et al. (2003) looks into the impact of board diversity on business value in the United States. The authors discover a link between a female director's presence and business performance as evaluated by Tobin's Q. Their conclusions sparked a succession of investigations into the evidence in several countries. However, the outcomes are still inconclusive.

### 3.1. Evidence from North America

The impact of demographic diversity on the board of directors is studied by Erhardt et al. (2003). Demographic diversity, the independent variable, is quantified in terms of ethnic and gender representation on boards. The authors discover a link between financial indices of firm success and demographic diversity on the board of directors in the United States. Francoeur et al. (2008) investigate the influence of gender diversity on corporate boards in firms with high betas, high market-to-book ratios, or a large standard deviation of analysts' estimates. According to the authors, having more women on boards of directors does not affect stock returns. Carter et al. (2010) investigate gender diversity on board committees and corporate boards in the United States. They discover no effect on business financial performance, both favourable and bad.

Miller and Del Carmen Triana (2009) look at how innovation and business reputation affect the relationship between board gender diversity and firm performance. There is no link between board gender diversity and business success, according to the authors. However, the findings reveal a link between board gender diversity and innovation (measured by R&D spending), but none with reputation. Adams and Ferreira (2009) examine the impact of gender diversity on governance measures and company performance in their important study. Women directors are less likely than men to have attendance issues, according to the authors. Female directors are also more likely than male directors to serve on monitoring committees. Women directors are more likely to be allocated to committees such as audit, nominating, and corporate governance. Furthermore, gender-diverse boards are more likely to hold CEOs accountable for bad stock price performance, according to the authors. Gender diversity on boards has a positive effect on companies with weak shareholder rights, whereas it is possible that more board oversight can improve company value but harms companies with strong shareholder rights.

### 3.2. Evidence from Europe, Australia, and Asia

According to Rose (2007), the presence of women on Danish boards has no bearing on firm performance as assessed by Tobin' Q. However, Campbell and Mínguez-Vera (2008) discover a favourable effect of the ratio of women on boards of directors on business value in Spain. Gender diversity initially harms company performance, but after reaching a "critical mass", it has a beneficial impact (Joecks et al. 2013). The small beneficial performance effect of board gender diversity disappears when the ratio of female directors hits a breakpoint of roughly 20%, according to Nguyen et al. (2015). Liu et al. (2014) found a favourable and substantial relationship between board gender diversity and business performance in a Chinese study. In the case of Chinese state-controlled businesses, however, the link is negligible. Women executive directors are more effective than women independent directors, according to the authors. Chapple and Humphrey (2014) evaluate the performance of portfolios of firms with gender-diverse Australian boards using an aggregate (market-level) approach and find no indication of a link between diversity and performance. There is, however, some evidence of a negative relationship between having several women on a board of directors and performance.

Low et al. (2015) investigated businesses in Hong Kong, South Korea, Malaysia, and Singapore. The authors conclude that increasing the number of female directors on the board has a beneficial effect on business performance after controlling for potential endogeneity between board gender diversity and firm performance. They also reveal that the attitude of the country regarding working women moderates this relationship. Abdullah et al. (2016) look at Malaysian companies to see how different performance indicators, ownership, and board structure affect performance. Gender diversity on boards has a favourable impact on accounting performance but has a detrimental impact on market performance, according to the authors. This relationship is strongly moderated by the kind of corporate ownership (government or family); government ownership is significant, while family ownership is small. Evidence from Europe, Australia, and Asia is likewise contradictory, implying that further research is needed in this area.

Many researchers studied the impact of board composition on another business performance measure, ROA. Previous research has shown that analysing this link produces mixed results, depending on the context and institutional contexts studied. Several studies, such as Sanan (2016), which was based in the Indian setting, have revealed a favourable association between gender diversity and a firm's financial performance as evaluated by ROA. Ward and Forker (2015) found that increased female representation on boards leads to better ROA in another study conducted in Ireland. Conyon and He (2017) discovered a link between a company's performance and the gender diversity of its board of directors. Carter et al. (2010), however, found no link between female presence on boards and ROA.

The above-mentioned discussion motivates the main hypothesis:

Companies that have at least a critical mass of female directors perform better than those that do not match this criterion.

## 4. Data Collection

The study aims to test the impact of gender diversity on firms' performance, besides testing the impact of a firm's internal characteristics on gender diversity. Therefore, the study uses the Return on Assets (ROA) and the Earning per Share (EPS) as accounting and financial indicators of firms' performance, respectively. Gender diversity reflects the women directors' share on board. The set of control variables includes CEO duality, the academic background of the board, firms' growth, size, age, and the financial leverage of the firm.

The most reliable source of information for data collection, yearly reports, were used to extract information on financial indicators. Earnings per share (EPS) and return on assets were the most popular and readily accessible profitability measurements in the Palestinian setting (ROA). All Palestinian businesses, across all industries, verified, consolidated, and uniform ROA and EPS as a result of Tobin's Q ratio inability to precisely quantify or forecast

the results of investments across certain time periods. Furthermore, a stock-performance metric could not be employed if it failed to identify a market or firm as being overpriced or undervalued (Brahma et al. 2021).

Return on assets (ROA), which measures how profitable a company is in relation to its assets and is used by investors to have a better knowledge of a company's financial performance and health, is the most crucial indicator for investors. ROA is calculated by dividing a company's net profit by all of its assets. Therefore, if a company's net profit is higher than a competitor's, it might claim that it is performing better. However, if the second company has a significantly higher ROA because it makes better use of its resources and capital, it may outperform the first one in the future. ROA measures the operating profit of firms. Compared with net income, ROA considers companies' size, which gives us a precise measure to compare the results (Brahma et al. 2021). The authors aimed to test for the impact of board composition on metrics other than ROA in order to make the results more reliable and less subject to how profitability is assessed. As a result, the EPS variable, which is derived by taking a company's net income and subtracting preferred dividends from it, was added to the model. Earnings per share is one of the most important indicators when seeking to assess a company's profitability on an absolute basis (EPS). This often-used proxied measure of corporate value reveals the earnings per share of a company. Since investors would pay more for a company's shares if they believe that earnings will increase in pace with the price of those shares, a higher EPS indicates a better value. A company's net profit was divided by the usual number of outstanding ordinary shares during the reporting year to arrive at earnings per share (EPS). As a result, we employed the fundamental EPS indicator and adhered to IAS 33. We subtracted that sum from the typical number of outstanding ordinary shares. These factors are frequently used as indicators of a company's performance (e.g., (Garanina and Muravyev 2021; Kramaric and Miletic 2017)).

Table 1 illustrate the dependent and independent variables, the abbreviations, and the way how these variables have been measured.

**Table 1.** The definition of each variable included in the models.

| Variable | Abbreviation | Measurement |
| --- | --- | --- |
| 1. Board characteristics (independent variables) | | |
| CEO duality | Duality | Dummy variable takes the value of 1 if the CEO is a board chair, otherwise 0. |
| Academic Background | Ph.D. | Percentage of board members with PhD qualification. |
| 2. Firm performance (dependent variable proxies) | | |
| Return on Assets | ROA | Net income divided by total assets. |
| Earnings per Share | EPS | Net Income divided by the number of shares outstanding. |
| 3. Control variables | | |
| Firm size | size | The logarithm of the firm's total assets in US Dollars (firms' data in Jordan Dinar is converted in US Dollars using the official average exchange rate) |
| Firm leverage | Leverage | Percentage of total liability to total assets |
| Growth opportunities | Growth | The price to book ratio has been considered as the proxy for growth opportunities. This is a ratio of the stock's market value to its book value |
| Firm age | age | Firm age |
| 4. Key independent variable | | |
| Gender diversity | Gender | Percentage of women directors on board. |

The study collected data from two main sources. The first source was the published data on Palestine Stock Exchange (PEX) website for the publicly traded firms. The second source was from the annual financial and managerial reports disclosed by the firms. Currently, PEX has 49 firms. The study differentiates between the financial firms (33) and the

non-financial (14), and eliminated two recently listed companies. The dataset covers the period 2010–2020.

## 5. Methodology

To investigate the impact of the internal determinants of gender diversity at the listed firms in Palestine exchange, this article employs the random panel data method as we do believe that the listed firms are heterogenous and the impact of each firm has a different slope.

$$Gender_{it} = \propto + \beta_1 PhD_{it} + \beta_2 growth_{it} + \beta_3 age_{it} + \beta_4 size_{it} + \beta_5 Duality_{it} + \mu_i + \lambda_t + \epsilon_{it} \quad (1)$$

where:

The outcome variable of this model is Gender diversity which represents the share of females in BoD at firm $i$ and time $t$.

$\propto$: is the constant term and is the same across all groups and time.

The firm's internal determinants of gender diversity are ($PhD$) which represents the share of PhD holders to the total number of BoD, ($growth$) reflects the firm's growth rate, the age of the firm ($age$), the size of the firm ($size$), and $Duality$.

$\mu_i$: captures any unobservable individual-specific effects

$\lambda_t$: captures any unobservable time-specific effects.

$\epsilon_{it}$: the error term, the model suggests using the Hubbards white standard error to avoid the problem of Heteroskedasticity.

Equation (1) tests the linear impact of these determinants on gender diversity. Nonetheless, to test the nonlinear impact of PhD holders and firm's growth rate on gender diversity, the study uses the following equations, respectively.

$$Gender_{it} = \propto + \beta_1 PhD_{it} + \gamma_1 PhD_{it}^2 + \beta_2 growth_{it} + \beta_3 age_{it} + \beta_4 size_{it} + \beta_5 Duality_{it} + \mu_i + \lambda_t + \epsilon_{it} \quad (2)$$

$$Gender_{it} = \propto + \beta_1 PhD_{it} + \beta_2 growth_{it} + \gamma_1 growth_{it}^2 + \beta_3 age_{it} + \beta_4 size_{it} + \beta_5 Duality_{it} + \mu_i + \lambda_t + \epsilon_{it} \quad (3)$$

However, to infer the linear impact of gender diversity on the firm's performance, the study uses the IV estimates to mitigate the endogeneity problem. The following equation reflects the preceded aim.

$$Y_{it} = \propto + \beta \sum_{1}^{n} X_{it} + \gamma \, gender_{it} + \mu_i + \lambda_t + \epsilon_{it} \quad (4)$$

$Y$: is the outcome variable that represents the performance of firm $i$ and time $t$, the study suggests four different proxies to indicate the firm's performance: Return on Assets and Earning per share.

$\propto$: is the constant term and is the same across all groups and time.

$X$: is the matrix of control variables that include the academic Background of the board of directors, CEO duality, firm size, firm leverage, firm's age, and firm's growth opportunities.

$\beta$: is a vector of control variables coefficients.

$gender$: the key independent variable of this research represents the gender diversity in firm $i$ and time $t$, the indicator represents the share of females on board of directors.

$\gamma$: is the coefficient of interest.

$\mu_i$: captures any unobservable individual-specific effects

$\lambda_t$: captures any unobservable time-specific effects.

$\epsilon_{it}$: the error term, the model suggests using the Hubbards white standard error to avoid the problem of Heteroskedasticity.

The study assumed that the financial leverage can be a function of ROA and EPS. Thus, reverse causality could arise here. The leverage in Equation (4) is instrumented by one lag of the following variables, leverage, firm's growth, size, and the year-specific fixed effect.

Moreover, to test whether a non-linear relationship exists between gender diversity and a firm's performance, the presence of a quadratic relationship demonstrates the existence

of an extreme point that reflects the optimal or worst share of females on the board of directors.

$$Y_{it} = \propto + \beta \sum_{1}^{n} X_{it} + \gamma \, gender_{it} + \delta \, (gender_{it})^2 + \mu_i + \lambda_t + \epsilon_{it} \tag{5}$$

The shape of the quadratic relationship creates the extreme point. This shape can be determined by the obtained signs $\gamma$ and $\delta$; both should be significant and in an opposite direction. Therefore, the extreme point can be estimated by using the first derivative as follows:

$$\frac{\partial Y}{\partial \, gender} = \gamma + \delta \, gender = 0$$

Therefore, the optimal share of females in the board of directors (inverse U-shape) is:

$$gender = \frac{-\gamma}{\delta}$$

The worst share of females in the board of directors (U-shape) is:

$$gender = \frac{-\gamma}{\delta}$$

## 6. Results and Discussion

As Table 2 shows, The return on assets in Palestine Exchange varied between ($-1.024$) to (0.261). The highest ratio was found in The Vegetable Oil Industries Co., Ltd. in 2013, whereas the lowest accounted for PalAqar for Real Estate Development & Management in 2014. According to the ROA, the homogeneity in the financial sector is more as the variation of ROA in the non-financial sector is 1.5 times higher than in the financial sector.

**Table 2.** Descriptive statistics.

| Variable | Obs. | Mean | Std. Dev. | Min | Max |
|---|---|---|---|---|---|
| **Full Sample** | | | | | |
| ROA | 465 | 0.024 | 0.077 | −1.024 | 0.261 |
| EPS | 467 | 0.168 | 0.345 | −1.691 | 2.408 |
| Female Ratio | 482 | 5.36 | 10.12 | 0 | 42.857 |
| PhD holders | 482 | 9.621 | 12.333 | 0 | 54.545 |
| Growth | 444 | 1.449 | 5.059 | 0.062 | 104.927 |
| leverage | 487 | 0.622 | 1.28 | 0.009 | 10.66 |
| Age | 505 | 23.804 | 14.618 | 0 | 75 |
| Size | 493 | 17.71 | 1.711 | 10.384 | 22.483 |
| Duality | 482 | 0.222 | 0.416 | 0 | 1 |
| **Financial Firms** | | | | | |
| ROA | 144 | 0.024 | 0.035 | −0.116 | 0.146 |
| EPS | 144 | 0.162 | 0.172 | −0.246 | 1.081 |
| Female Ratio | 140 | 4.804 | 8.11 | 0 | 33.333 |
| PhD holders | 140 | 11.43 | 12.146 | 0 | 54.545 |
| Growth | 137 | 2.067 | 9.005 | 0.081 | 104.927 |
| leverage | 143 | 1.347 | 2.183 | 0.271 | 10.66 |
| Age | 148 | 19.986 | 12.142 | 0 | 60 |
| Size | 145 | 18.742 | 1.804 | 10.384 | 22.483 |
| Duality | 140 | 0.307 | 0.463 | 0 | 1 |
| **Non-financial Firms** | | | | | |
| ROA | 321 | 0.025 | 0.089 | −1.024 | 0.261 |
| EPS | 323 | 0.171 | 0.398 | −1.691 | 2.408 |
| Female Ratio | 342 | 5.587 | 10.839 | 0 | 42.857 |
| PhD holders | 342 | 8.881 | 12.35 | 0 | 42.857 |
| Growth | 307 | 1.174 | 0.877 | 0.062 | 5.723 |
| leverage | 344 | 0.321 | 0.194 | 0.009 | 1 |
| Age | 357 | 25.387 | 15.267 | 0 | 75 |
| Size | 348 | 17.28 | 1.472 | 13.562 | 21.087 |
| Duality | 342 | 0.187 | 0.391 | 0 | 1 |

**Table 3.** Matrix of correlations.

| Variables | (1) | (2) | (3) | (4) | (5) | (6) | (7) | (8) | (9) |
|---|---|---|---|---|---|---|---|---|---|
| (1) ROA | 1.000 | | | | | | | | |
| (2) EPS | 0.711 | 1.000 | | | | | | | |
| (3) Female Ratio | 0.092 | 0.032 | 1.000 | | | | | | |
| (4) PhD holders | 0.123 | −0.001 | 0.157 | 1.000 | | | | | |
| (5) Growth | −0.048 | 0.010 | 0.054 | −0.150 | 1.000 | | | | |
| (6) Leverage | −0.182 | −0.108 | −0.011 | −0.097 | 0.202 | 1.000 | | | |
| (7) Age | 0.243 | 0.392 | 0.044 | −0.011 | −0.106 | −0.270 | 1.000 | | |
| (8) Size | 0.171 | 0.206 | −0.231 | 0.055 | −0.127 | 0.280 | −0.038 | 1.000 | |
| (9) Duality | 0.012 | −0.053 | −0.166 | −0.100 | −0.120 | 0.169 | 0.287 | 0.087 | 1.000 |

The earnings per share fluctuated between (−1.691) PalAqar for Real Estate Development & Management in 2014 and (2.408) The Vegetable Oil Industries Co., Ltd. in 2013. Again, the dispersion of EPS in the non-financial sector is larger than in the financial one. Table 3 presents the correlation matrix. It shows a high association between ROA and EPS which means that as ROA changes in a specific direction, the EPS most likely tends to change in the same direction.

However, the female representation at the listed firms in PEX varied between zero percentage to (42.86). In the financial sector, the highest ratio was recorded at the Bank of Palestine in 2019 and 2020, while in the non-financial sector, it was recorded at Palestine Investment and Development during the years 2011–2020. Figure 1 shows the average number of female directors in the board of directors in the sull sample, financial, and non-financial firms during the years 2010–2020.

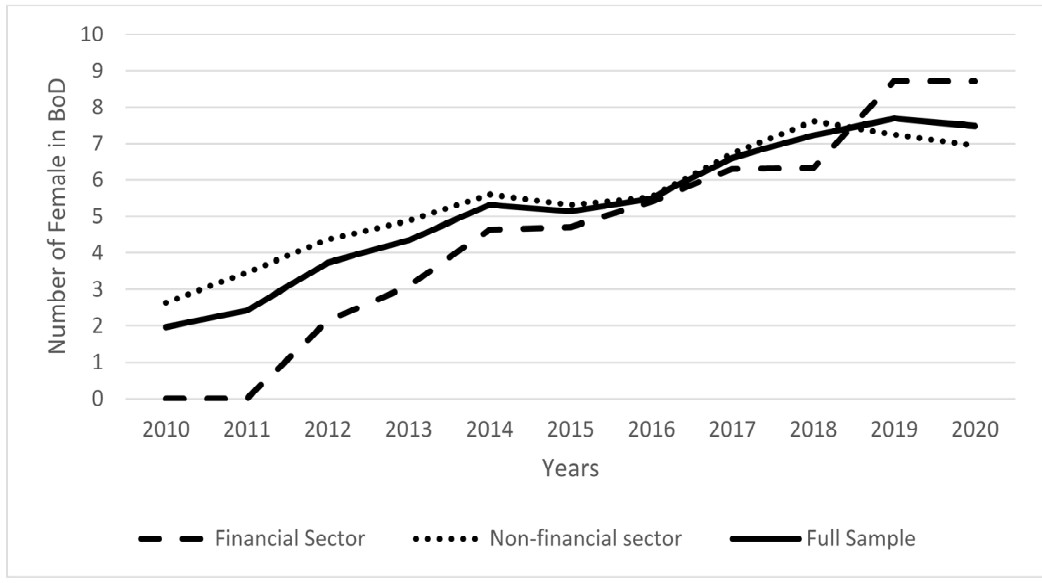

**Figure 1.** The average number of female directors in the board of directors in the sull sample, financial, and non-financial firms during the years 2010–2020.

Table 4 shows the impact of a firm's internal determinants such as the share of PhD holders, firm's age, size, growth, and the duality of the board of directors and CEO following Equation (1). Model (1) shows the results of the full sample and reveals that duality and firm growth harm gender diversity. However, the age of the firm has a positive impact at %10. Model (2) and Model (3) present the results for financial and non-financial firms, respectively. In Model (2) the existence of PhD holders increases at %10 level of significance the representation of the female in BOD, whereas the other results are robust with Model (1). Nevertheless, the results failed to conclude any significant results with the same model specifications in the non-financial firms.

**Table 4.** Gender Diversity determinants at the Palestinian listed firms.

| | (1) | (2) | (3) |
|---|---|---|---|
| VARIABLES | Full Sample | Financial Firms | Non-Financial Firms |
| PhD holders | 0.180 | 0.329 * | 0.112 |
| | (0.206) | (0.183) | (0.323) |
| Firm growth | −0.116 *** | −0.125 *** | 0.780 |
| | (0.0426) | (0.0443) | (1.097) |
| Firm age | 0.0670 * | 0.496 *** | 0.0937 |
| | (0.0346) | (0.156) | (0.0617) |
| Firm size | 0.482 | −1.307 | 1.000 |
| | (1.399) | (1.554) | (2.063) |
| Duality | −4.478 ** | −10.37 *** | −1.018 |
| | (2.169) | (2.417) | (0.931) |
| Constant | −12.32 | 20.45 | −22.69 |
| | (23.03) | (24.47) | (35.16) |
| Observations | 436 | 134 | 302 |
| Number of groups | 47 | 14 | 33 |
| Firm fixed effect | Yes | Yes | Yes |
| Year fixed effect | Yes | Yes | Yes |
| Robust | Yes | Yes | Yes |
| Country clustering | Yes | Yes | Yes |

Robust standard errors in parentheses *** $p < 0.01$, ** $p < 0.05$, * $p < 0.1$.

For more in-depth analysis, the study aimed to test the non-linear relationship between the level of PhD certificate holders on BOD and gender diversity. All models in Table 5 concluded that the relationship is quadratic, and the parabola has an inverted U-shape. One can get the optimal PhD in BOD that empowers the highest share of females by taking the first derivative of Equation (2) and equalling it by zero, the following equation presenting the preceded discussion for the full sample depending on the results obtained from Table 5, Model (1)

$$\frac{\partial\ Female\ Ratio}{\partial\ Ph.D.} = 0.712 - 0.0139\ Ph.D. = 0$$

$$The\ optimal\ Ph.D.\ \cong \%51$$

This percentage is less at the non-financial firms almost by %10. However, this percentage should increase up to %81 in financial firms.

Results in Table 6 show the non-linear impact of a firm's growth on gender diversity in BOD following Equation (3). Model (1) presents the results of the full sample and concluded that a quadratic relationship does exist, the parabola of this non-linear relationship takes the U-shape.

This means a firm must avoid a threshold equal to %121 of growth to keep the females' representation in the BOD away from its minimum. This relationship exists for the financial firms and approximately has a similar ratio. However, it does not take place in non-financial firms.

The study commences its analysis to investigate the impact of gender diversity on a firm's performance by employing linear models. Table 7 follows Equation (4) and shows the linear impact of gender diversity in BOD on ROA and EPS. All models failed to conclude a significant impact of gender diversity on neither ROA nor EPS. Moreover, the impact does not exist at any subsample. The study extends the analysis by inferring the non-linear impact of female representation on BOD.



**Table 5.** The non-linear impact of holding PhD certificates on gender diversity in BOD.

| | (1) | (2) | (3) |
|---|---|---|---|
| **VARIABLES** | **Full Sample** | **Financial Firms** | **Non-Financial Firms** |
| PhD | 0.712 ** | 0.639 ** | 1.190 * |
| | (0.301) | (0.269) | (0.610) |
| PhD * PhD | −0.0139 ** | −0.00785 ** | −0.0285 ** |
| | (0.00648) | (0.00354) | (0.0137) |
| Firm growth | −0.0984 ** | −0.115 *** | 0.778 |
| | (0.0407) | (0.0425) | (1.123) |
| Firm age | 0.0662 * | 0.432 *** | 0.0898 |
| | (0.0340) | (0.144) | (0.0624) |
| Firm size | 0.568 | −1.377 | 0.891 |
| | (1.208) | (1.439) | (2.194) |
| Duality | −3.642 ** | −8.626 *** | −0.938 |
| | (1.579) | (2.743) | (0.851) |
| Constant | −13.65 | 21.16 | −20.68 |
| | (19.76) | (22.24) | (37.46) |
| Observations | 436 | 134 | 302 |
| Number of Panels | 47 | 14 | 33 |
| Firm fixed effect | Yes | Yes | Yes |
| Year fixed effect | Yes | Yes | Yes |
| Robust | Yes | Yes | Yes |
| Country clustering | Yes | Yes | Yes |

Robust standard errors in parentheses *** $p < 0.01$, ** $p < 0.05$, * $p < 0.1$.

**Table 6.** The non-linear impact of firm's growth on gender diversity in BOD.

| | (1) | (2) | (3) |
|---|---|---|---|
| **VARIABLES** | **Full Sample** | **Financial Firms** | **Non-Financial Firms** |
| PhD | 0.181 | 0.356 * | 0.117 |
| | (0.205) | (0.185) | (0.328) |
| Firm growth | −0.786 ** | −1.006 *** | 3.507 |
| | (0.333) | (0.295) | (3.337) |
| Firm (growth * growth) | 0.00648 ** | 0.00846 *** | −0.510 |
| | (0.00314) | (0.00283) | (0.503) |
| Firm age | 0.0417 | 0.466 *** | 0.149 |
| | (0.0378) | (0.162) | (0.0967) |
| Firm size | 0.321 | −1.071 | 0.655 |
| | (1.408) | (1.607) | (2.000) |
| Duality | −4.367 ** | −9.976 *** | −1.071 |
| | (2.128) | (2.341) | (0.949) |
| Constant | −8.732 | 17.83 | −19.81 |
| | (23.21) | (25.11) | (34.00) |
| Observations | 436 | 134 | 302 |
| Number of panels | 47 | 14 | 33 |
| Firm fixed effect | Yes | Yes | Yes |
| Year fixed effect | Yes | Yes | Yes |
| Robust | Yes | Yes | Yes |
| Country clustering | Yes | Yes | Yes |

Robust standard errors in parentheses *** $p < 0.01$, ** $p < 0.05$, * $p < 0.1$.

**Table 7.** The linear impact of gender diversity in BOD on ROA and EPS.

| VARIABLES | (1) Full Sample | (2) Financial Firms | (3) Non-Financial Firms | (4) Full Sample | (5) Financial Firms | (6) Non-Financial Firms |
|---|---|---|---|---|---|---|
| | ROA | ROA | ROA | EPS | EPS | EPS |
| Leverage | −0.0123 | 0.00998 *** | 0.0804 | −0.0135 | 0.0131 | −0.281 |
| | (0.00893) | (0.00256) | (0.135) | (0.0204) | (0.0178) | (0.282) |
| Female ratio | 0.000225 | −0.000403 | 0.000359 | −0.00255 | −0.00421 | 0.000490 |
| | (0.000514) | (0.000343) | (0.000818) | (0.00178) | (0.00309) | (0.00209) |
| Size | −0.0512 *** | $-2.09 \times 10^{-5}$ | −0.0564 | 0.0468 | 0.0816 | 0.278 * |
| | (0.0150) | (0.00849) | (0.0383) | (0.0599) | (0.0546) | (0.154) |
| Age | 0.00365 ** | −0.000899 | 0.00124 | 0.00308 | −0.00485 | −0.00534 |
| | (0.00149) | (0.00131) | (0.00168) | (0.00562) | (0.0116) | (0.00593) |
| PhD | 0.00358 ** | 0.000478 | 0.00482 ** | 0.00662 * | 0.00422 | 0.00848 ** |
| | (0.00173) | (0.000466) | (0.00194) | (0.00341) | (0.00374) | (0.00332) |
| Growth | −0.0139 ** | −0.0161 *** | −0.0175 | −0.0588 ** | −0.0792 ** | 0.0324 |
| | (0.00668) | (0.00379) | (0.0246) | (0.0235) | (0.0316) | (0.0565) |
| Constant | 0.834 *** | 0.0441 | 0.919 | −0.716 | −1.268 | −4.516 * |
| | (0.241) | (0.140) | (0.623) | (0.989) | (0.839) | (2.572) |
| Observations | 385 | 120 | 265 | 385 | 120 | 265 |
| Number of panels | 47 | 14 | 33 | 47 | 14 | 33 |
| Year fixed effect | Yes | Yes | Yes | Yes | Yes | Yes |
| Robust | Yes | Yes | Yes | Yes | Yes | Yes |
| Firm clustering | Yes | Yes | Yes | Yes | Yes | Yes |

Robust standard errors in parentheses *** $p < 0.01$, ** $p < 0.05$, * $p < 0.1$.

Model (1) and Model (4) in Table 8 demonstrate that the impact of gender diversity on a firm's performance for the full sample can be represented by a quadratic curve that has a U-shape parabola. This implies that females in BOD of the Palestinian firms harm performance until reaching a specific ratio where the impact starts to be inverted. The results of Table 1 suggested that the lowest ROA takes place when 29.4% of BOD are females, see Figure 2. Before this ratio, the impact is negative and starts to be positive afterwards. However, this ratio increases in the case of EPS to 41.1%, see Figure 3. The non-financial firms were the channel of significance on the full sample as their results are consistent with the preceded findings see Figures 4 and 5.

The results appear to be in line with empirical studies by Kanter (1977), Kramer et al. (2006), Kristie (2011), Catalyst (2013), Strydom et al. (2016), Brahma et al. (2021), Garanina and Muravyev (2021), and Carmo et al. (2022), as well as the critical mass theory, which postulates that while women may have been a minority or token before reaching the critical, the return on assets (ROA) and earnings per share are typically used in these studies to evaluate the financial performance (EPS). Our results were at odds with ideas that contend that there is a direct correlation between gender diversity and financial performance. First, the human capital hypothesis (Becker 1985) contends that a diverse board results in a pool of collective talents, experience, and education that improves business performance. Our findings show that below the critical mass, the human capital hypothesis has little bearing. Second, the results appear to be at odds with the social identity theory (Tajfel and Turner 1979) that suggests a linear relationship because, as the proportion of female directors rises, the homogeneity in decision-making becomes obvious, undermining the efforts of the male-dominated board to exclude the minority role of women.

**Table 8.** The non-linear impact of gender diversity in BOD on ROA and EPS.

| | (1) | (2) | (3) | (4) | (5) | (6) |
|---|---|---|---|---|---|---|
| VARIABLES | Full Sample | Financial Firms | Non-Financial Firms | Full Sample | Financial Firms | Non-Financial Firms |
| | ROA | ROA | ROA | EPS | EPS | EPS |
| Leverage | −0.00945 | 0.00993 *** | 0.0917 | −0.00703 | 0.0129 | −0.259 |
| | (0.00785) | (0.00259) | (0.132) | (0.0186) | (0.0165) | (0.275) |
| Female ratio | −0.00350 * | $-8.15 \times 10^{-5}$ | −0.00545 * | −0.0108 *** | 0.000787 | −0.0122 ** |
| | (0.00203) | (0.000828) | (0.00329) | (0.00372) | (0.00389) | (0.00597) |
| (Female ratio)$^2$ | 0.000119 * | $-1.19 \times 10^{-5}$ | 0.000177 * | 0.000263 ** | −0.000185 | 0.000387 ** |
| | $(6.77 \times 10^{-5})$ | $(2.86 \times 10^{-5})$ | $(9.20 \times 10^{-5})$ | (0.000125) | (0.000153) | (0.000157) |
| Size | −0.0509 *** | 0.000459 | −0.0584 | 0.0476 | 0.0907 * | 0.275 * |
| | (0.0146) | (0.00875) | (0.0380) | (0.0591) | (0.0531) | (0.151) |
| Age | 0.00319 ** | −0.000981 | −0.000127 | 0.00204 | −0.00637 | −0.00833 |
| | (0.00144) | (0.00136) | (0.00193) | (0.00539) | (0.0117) | (0.00589) |
| PhD | 0.00364 ** | 0.000501 | 0.00489 *** | 0.00675 ** | 0.00458 | 0.00862 *** |
| | (0.00171) | (0.000487) | (0.00189) | (0.00334) | (0.00381) | (0.00324) |
| Growth | −0.0141 ** | −0.0162 *** | −0.0202 | −0.0592 ** | −0.0808 ** | 0.0269 |
| | (0.00657) | (0.00378) | (0.0245) | (0.0235) | (0.0316) | (0.0555) |
| Constant | 0.846 *** | 0.0362 | 1.003 | −0.692 | −1.416 * | −4.342 * |
| | (0.240) | (0.144) | (0.641) | (0.979) | (0.812) | (2.525) |
| Observations | 385 | 120 | 265 | 385 | 120 | 265 |
| Number of Panels | 47 | 14 | 33 | 47 | 14 | 33 |
| Robust | Yes | Yes | Yes | Yes | Yes | Yes |
| Year fixed effect | Yes | Yes | Yes | Yes | Yes | Yes |
| Firm clustering | Yes | Yes | Yes | Yes | Yes | Yes |

Robust standard errors in parentheses *** $p < 0.01$, ** $p < 0.05$, * $p < 0.1$.

The findings provide some support for the resource dependence theory, as do those by Pfeffer and Salancik (1978), Cox et al. (1991), Kravitz (2003), Huse and Solberg (2006), and Hillman et al. (2007). Since women have a voice, legitimacy, and communication channels with other bodies, their presence on the board of directors lessens the impact of dependency. This is not the case when there is a low percentage of women or when the board is controlled by men. Our findings, however, show a non-linear relationship that is not supported by this idea. The argument is that because women's rights are still not widely accepted in Palestinian culture, corporations cannot fully legitimize themselves by choosing female directors, especially when their representation is minimal. Finally, the agency theory research provides some support for our findings. According to Fama and Jensen (1983), Gul et al. (2008), and Adams and Ferreira (2009), women only become more advanced in monitoring activities, auditing, and management accounting, which improves firm performance. However, our study is only correct once they have reached a critical mass.

The study has conducted further analysis by using quantile regression analysis, results in Table 9 shows the quantile impact of gender diversity on ROA; However, Table 10 shows the quantile impact of gender diversity on EPS.

**Table 9.** The Quantile impact of gender diversity on ROA.

|  | (1) | (2) | (3) | (4) | (5) | (6) |
|---|---|---|---|---|---|---|
| **VARIABLES** | **OLS** | **0.25** | **0.5** | **0.75** | **0.9** | **0.95** |
| Leverage | 0.00484 * | 0.00845 *** | 0.00830 *** | 0.00601 ** | 0.00167 | −0.000681 |
|  | (0.00273) | (0.00208) | (0.00172) | (0.00296) | (0.00396) | (0.00407) |
| Female ratio | 0.000542 | $3.54 \times 10^{-5}$ | 0.000219 | 0.000357 | 0.00195 *** | 0.00218 *** |
|  | (0.000357) | (0.000271) | (0.000225) | (0.000387) | (0.000517) | (0.000532) |
| Size | 0.00305 | −0.000272 | −0.00326 ** | −0.00521 ** | −0.000253 | 0.00793 ** |
|  | (0.00229) | (0.00174) | (0.00144) | (0.00249) | (0.00333) | (0.00342) |
| Age | 0.00117 *** | 0.000395 ** | 0.000382 ** | 0.00166 *** | 0.00179 *** | 0.00222 *** |
|  | (0.000249) | (0.000189) | (0.000157) | (0.000270) | (0.000361) | (0.000371) |
| PhD | 0.000408 | 0.000417 * | $9.05 \times 10^{-5}$ | −0.000227 | −0.000854 * | −0.00108 ** |
|  | (0.000307) | (0.000233) | (0.000193) | (0.000333) | (0.000445) | (0.000457) |
| Growth | $-5.22 \times 10^{-5}$ | $8.63 \times 10^{-5}$ | $-6.24 \times 10^{-5}$ | −0.000191 | −0.000589 | −0.000826 |
|  | (0.000722) | (0.000549) | (0.000455) | (0.000783) | (0.00105) | (0.00108) |
| Constant | −0.0669 * | −0.0113 | 0.0622 ** | 0.108 ** | 0.0568 | −0.0771 |
|  | (0.0406) | (0.0308) | (0.0255) | (0.0440) | (0.0588) | (0.0604) |
| Observations | 430 | 430 | 430 | 430 | 430 | 430 |
| R-squared | 0.077 |  |  |  |  |  |

Robust standard errors in parentheses *** $p < 0.01$, ** $p < 0.05$, * $p < 0.1$.

**Table 10.** The Quantile impact of gender diversity on EPS.

|  | (1) | (2) | (3) | (4) | (5) | (6) |
|---|---|---|---|---|---|---|
| **VARIABLES** | **OLS** | **0.25** | **0.5** | **0.75** | **0.9** | **0.95** |
| Leverage | −0.0126 | 0.00833 | 0.00344 | −0.00943 | −0.0393 | −0.0680 ** |
|  | (0.0117) | (0.00517) | (0.00741) | (0.0155) | (0.0494) | (0.0331) |
| Female ratio | 0.00102 | 0.000153 | 0.000378 | 0.00231 | 0.00640 | 0.00820 * |
|  | (0.00153) | (0.000676) | (0.000967) | (0.00203) | (0.00645) | (0.00432) |
| Size | 0.0284 *** | 0.0146 *** | 0.0185 *** | 0.0418 *** | 0.0766 * | 0.0891 *** |
|  | (0.00985) | (0.00434) | (0.00621) | (0.0130) | (0.0415) | (0.0277) |
| Age | 0.00893 *** | 0.00252 *** | 0.00389 *** | 0.00717 *** | 0.0193 *** | 0.0297 *** |
|  | (0.00107) | (0.000471) | (0.000674) | (0.00141) | (0.00450) | (0.00301) |
| PhD | −0.00115 | 0.00200 *** | 0.00106 | −0.00241 | −0.00728 | −0.00472 |
|  | (0.00132) | (0.000581) | (0.000832) | (0.00174) | (0.00555) | (0.00371) |
| Growth | 0.000202 | 0.000321 | −0.000207 | −0.00156 | −0.00292 | −0.00361 |
|  | (0.00310) | (0.00137) | (0.00196) | (0.00411) | (0.0131) | (0.00874) |
| Constant | −0.534 *** | −0.324 *** | −0.348 *** | −0.665 *** | −1.242 * | −1.447 *** |
|  | (0.174) | (0.0768) | (0.110) | (0.231) | (0.734) | (0.491) |
| Observations | 430 | 430 | 430 | 430 | 430 | 430 |
| R-squared | 0.164 |  |  |  |  |  |

Robust standard errors in parentheses *** $p < 0.01$, ** $p < 0.05$, * $p < 0.1$.

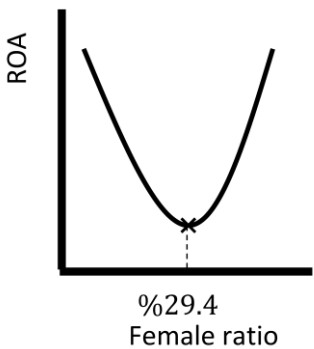

**Figure 2.** The critical female ratio in the full sample with respect to ROA.

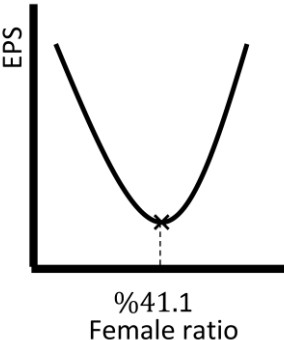

**Figure 3.** The critical female ratio in the full sample with respect to EPS.

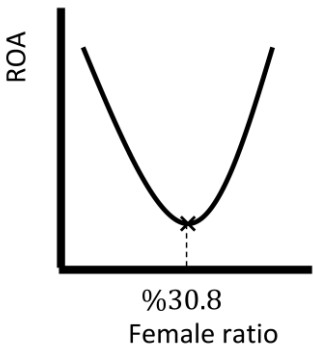

**Figure 4.** The critical female ratio in non-financial firms with respect to ROA.

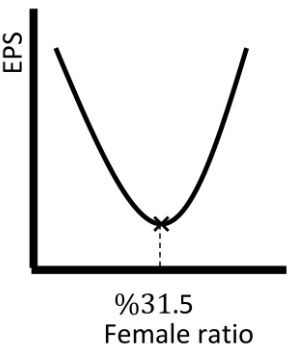

**Figure 5.** The critical female ratio in non-financial firms with respect to EPS.

## 7. Conclusions

The study aimed to test the impact of gender diversity on a firm's performance. Namely, the non−linear impact at the listed companies in Palestine Exchange during the period from 2010 to 2020, besides testing the impact of firm's internal characteristics on gender diversity. To that end, the study utilised the Return on Assets (ROA) and the Earning per Share (EPS) as accounting and financial indicators of firms' performance, respectively, whereas gender diversity reflects the women directors share on board. The study used the instrument analysis to test the impact of gender diversity on firms' performance. The results were consistent with the critical mass theory and the token status theory as we found that the impacts are non−linear. In addition, the quantile analysis suggested that the impact of female directors on boards becomes greater in firms that achieve the highest ROA, see Figure 6 and EPS, see Figure 7.

As mentioned previously, the main contributions of this study are three-fold. Firstly, to test the impact of female representation in PEX bord of directors, the investigations include the linear and the existence of the critical mass relationship. Secondly, to determine the impact of firms' internal characteristics including holding PhD's, firms' size, growth, and age; besides, including governance indicators namely duality on gender diversity

in PEX. Thirdly, to test at which level of output the impact of gender diversity on firms' performance is more effective. This could be tested by conducting the quantile regression.

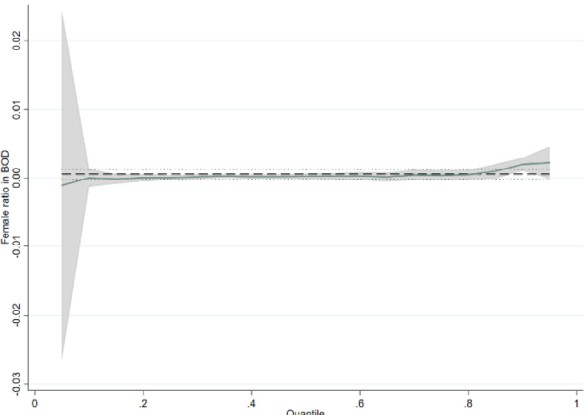

**Figure 6.** The quartile impact of female ratio on ROA.

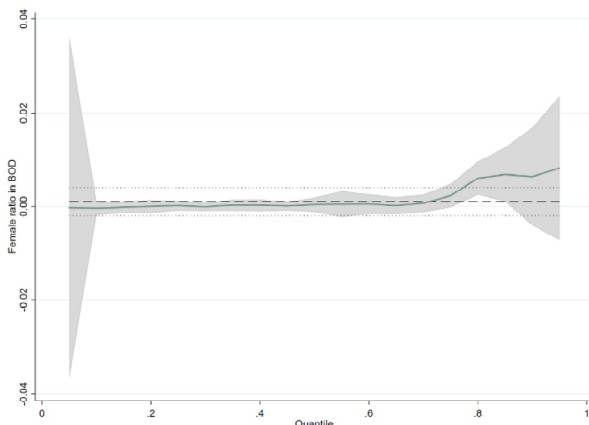

**Figure 7.** The quartile impact of female ratio on EPS.

The policy implication of this study can be summarised in the following points. First, although financial firms are empowering more females in BOD, they still have to increase their representation by more than 30%. Second, the highest ROA and EPS companies are recommended to increase the presence of a female on board as they can impact the performance much better. Third, firms need to empower more PhD holders to support more diversified boards. Future studies are recommended to test the channels that can better support the existence of females in BOD such as education and social responsibility. The main limitations of this paper were the data availability. We spent time on collecting the data from different sources including the annual financial and managerial reports, and part of the data was not verified by the PEX. The other limitation was the time span. The study collected data for the period 2010–2020, and data after the year 2020 and before the year 2010 is not available.

**Author Contributions:** Conceptualization, M.Q.; methodology, A.J.K.A.; software, A.J.K.A.; validation, M.E.; formal analysis, A.J.K.A.; investigation, A.J.K.A. and N.A.; resources, N.A.; data curation, N.A.; writing—original draft preparation, A.J.K.A., M.Q. and N.A.; writing—review and editing, A.J.K.A. and M.Q.; visualization, M.E. and N.A.; supervision, A.J.K.A. and M.E.; project administration, A.J.K.A.; funding acquisition (not applicable). All authors have read and agreed to the published version of the manuscript.

**Funding:** This research received no external funding.

**Data Availability Statement:** Data is available upon request, please contact the correspondent author.

**Conflicts of Interest:** The authors declare no conflict of interest.

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
