# Peer review of "The Internal Determinants of Gender Diversity and Its Non-Linear Impact on Firms’ Performance: Evidence from the Listed Companies in Palestine Exchange"

_jrfm, doi:10.3390/jrfm16010028_

Round 1

Reviewer 1 Report

This paper intends to explore the impact of gender diversity on a firm’s performance. The authors used the instrument analysis, traditional panel models, and quantile regression to fulfil the aims. The results demonstrate the existence of a critical mass for the impact of gender diversity on firms’ performance and that mass is about 30% for the ROA and 41% for the EPS.

The following are some suggestions for improvement:

·        The section of the literature is adequately developed, but I was expected to see some working hypotheses formulated, while these hypotheses might represent the starting point of the study

·        The authors said at line 265-268: “The study aims to test the impact of gender diversity on firms’ performance, besides testing the impact of a firm’s internal characteristics on gender diversity. Therefore, the study uses the Return on Assets (ROA) and the Earning per Share (EPS) are accounting and financial indicators of firms’ performance respectively”. My question is why these indicators and not others. Why are these indicators relevant? The authors could provide some arguments in justifying the selection of these indicators

·        The sentence from line 272 seems to unfinished: “The study uses the firms’ financial reports for collecting the data,…. “

·        I was expected to see in the section Data collection, how was the sample selected? How many companies, for what period? A more transparency in indicating the selection of the data would be recommended.

·        The section Results and discussion starts directly with a Table, it does not look so good. Try to insert an introductive paragraph before disclosing the Table 2.

·        At line 341, 347, 352, 364, etc   there are some Error! Reference source not found

·        The presentation of the paper should be improved. See for instance line 376 where is a sentence: Results in …

·        The discussion of the results needs more comparison with previous studies. How are the results similar to or different from what has been done before in related papers? This will help highlight any unique findings.

I wish you good luck with your paper!

Author Response

Dear respected reviewer,

Thanks for your time and efforts in providing these valuable comments which indeed enrich the submitted article.

On behalf of the team, we wish you a happy new year.

Kind regards,

Abdelrahman

Reviewer 2 Report

I thank the Journal for being able to read and review the paper which I found interesting.

In particular, while no new concepts are presented, the application of an established framework to Palestine market increases knowledge about the impact of women's presence on boards.

My suggestions to the authors, before the paper can be published, are as follows:

- check that the most recent literature on the topic has been cited;

- the theoretical framework referred to (i.e., token theory) should be made explicit, including reporting more empirical studies that are based on this framework;

- more information should be given about the market analyzed and the reference sample. Indeed, the innovativeness of the paper relates specifically to Palestinian firms;

- in the conclusions authors should  more extensively explain the contributions of their empirical work;

- always in the conclusions better explain the limitations of the survey;

- finally, re-read the text well to avoid typos.

Author Response

(The authors gave the same response as above.)

Round 2

Reviewer 1 Report

I believe the manuscript has been sufficiently improved to warrant publication in JRFM.